# Inequalities in Violent Death across Income Levels among Young Males and Females in Countries of the Americas

**DOI:** 10.3390/ijerph20075256

**Published:** 2023-03-24

**Authors:** Oscar J. Mujica, Dihui Zhang, Yi Hu, Isabel C. Espinosa, Nelson Araneda, Anca Dragomir, George Luta, Antonio Sanhueza

**Affiliations:** 1Pan American Health Organization, Washington, DC 20037, USA; 2Department of Biostatistics, Bioinformatics and Biomathematics, Georgetown University, Washington, DC 20057, USA; 3T.H. Chan School of Public Health, Harvard University, Boston, MA 02115, USA; 4Department of Education, University of La Frontera, Temuco 4811230, Chile; 5Department of Oncology, Georgetown University, Washington, DC 20057, USA

**Keywords:** road traffic injuries, homicide, suicide, mortality, heath status disparities, gender, social determinants of health, health equity, the Americas

## Abstract

Background: Violent deaths (i.e., those due to road traffic injury, homicide, and suicide) are among the most important causes of premature and preventable mortality in young people. This study aimed at exploring inequalities in violent death across income levels between males and females aged 10 to 24 years from the Americas in 2015, the SDG baseline year. Methods: In a cross-sectional ecological study design, eleven standard summary measures of health inequality were calculated separately for males and females and for each cause of violent death, using age-adjusted mortality rates and average income per capita for 17 countries, which accounted for 87.9% of the target population. Results: Premature mortality due to road traffic injury and homicide showed a pro-poor inequality pattern, whereas premature mortality due to suicide showed a pro-rich inequality pattern. These inequalities were statistically significant (*p* < 0.001), particularly concentrated among young males, and dominated by homicide. The ample array of summary measures of health inequality tended to generate convergent results. Conclusions: Significant inequalities in violent death among young people seems to be in place across countries of the Americas, and they seem to be socially determined by both income and gender. These findings shed light on the epidemiology of violent death in young people and can inform priorities for regional public health action. However, further investigation is needed to confirm inequality patterns and to explore underlying mechanisms, age- and sex-specific vulnerabilities, and gender-based drivers of such inequalities.

## 1. Introduction

The year 2015 marked the end of the Millennium Development Goals era. For the region of the Americas, this era brought much to celebrate, including economic growth [1], increases in national educational attainment for both men and women, and the expansion of health services for millions of people. Despite these achievements, in some areas of development young people aged 10 to 24 years continued to be at a heightened risk of dying prematurely from preventable causes, as reflected by their increasing mortality rates between 2000 and 2015 [2]. Among these causes of premature, preventable mortality among young people, violent deaths due to road traffic injuries, homicide, and suicide stand out [3].

The year 2015 also marked the beginning of the Sustainable Development Goals (SDG) agenda [4]. The region of the Americas has embraced this new global health agenda that prioritizes universal health and equity, as well as the new Global Strategy for Women’s, Children’s and Adolescents’ Health [5]. The SDG Agenda explicitly defines global target indicators for road traffic injury (3.6.1), homicide (16.1.1), and suicide (3.4.2). For the countries from this region to implement this ambitious agenda and fulfill its promise to leave no one behind, they have to start looking beyond regional and national averages and examine inequalities at the local level [6]. In a region dubiously distinguished by being recognized as the most inequitable in the world, especially in terms of income inequality, [1,6] there is an urgent need to identify groups that are disproportionately affected by poor health and/or groups that are at an increased risk of dying prematurely due to avoidable causes, and thereafter act on their social determinants, including the gender-related drivers of these inequalities. To do so, disaggregated data need to be analyzed repeatedly using equity-based measurements.

To support countries in this monitoring task, this study aims at defining a baseline by exploring inequalities in mortality due to the top three causes of violent death (i.e., road traffic injuries, homicide, and suicide) across income levels among young males and females aged 10–24 years, from selected countries of the Americas in 2015, the SDG baseline year.

## 2. Materials and Methods

A cross-sectional ecological study was conducted by using 2015 mortality data from countries of the Americas, available through the Pan American Health Organization (PAHO) Regional Mortality Database. To assemble this data set, death entries of all 34 countries and territories with 2015 data available in the regional mortality database at the time of the study were considered and reviewed for quality, based on data completeness and consistency. An additional standard validation process [7] was applied to correct for deaths that might not have been registered (i.e., under-registration), as well as for those with undetermined underlying causes of death (International Classification of Diseases, Tenth Revision, ICD-10, codes Y10–Y34), followed by proportional redistribution into sex, age group and main burden of disease categories (i.e., communicable, non-communicable, and injuries). ICD-10 codes used were V01–V89 (road traffic injury), X85–Y09 (homicide), and X60–X84 (suicide).

Age-adjusted, cause- and sex-specific mortality rates were calculated using the standard population distribution of the WHO world population age structure for the period 2000–2025. All rates were calculated per 100,000 population, as follows:(1)$$age\;adjusted\;rate=\sum_{i=1}^{3}\left[w_{i}\times \left(\frac{count_{i}}{population_{i}}\right)\right]\times 100,000$$
where $count_{i}$ is the number of deaths in age group i, $population_{i}$ is the population size of the age group *i*, and $w_{i}$ represents the weight of the age group *i* (derived from the standard population distribution, where the population percentages have been rescaled to add up to 1). The actual weights used for the 3 age groups (ages 10–14, 15–19, and 20–24) were 34.0%, 33.5% and 32.5%, respectively.

Standard summary measures of inequality in mortality due to road traffic injury, homicide, and suicide were calculated across the country hierarchy defined by income. We used 2015 estimates of gross national income per capita adjusted by purchase power in 2017-constant international dollars, as reported in the World Bank’s World Development Indicators database. Inequality measurements were computed using the Health Disparities Calculator (HD*Calc) [8], free statistical software developed by the US National Cancer Institute that calculates a set of 11 summary measures of health inequality, including range difference (RD), between-group variation (BGV), absolute concentration index (ACI), slope index of inequality (SII), range ratio (RR), index of disparity (IDisp), mean log deviation (MLD), relative concentration index (RCI), Theil index (T), relative index of inequality (RII), and the Kunst–Mackenbach relative index (KMI) [9,10,11]. Four of these quantities (RD, BGV, ACI, and SII) measure absolute inequality, while the remaining seven measure relative inequality. It is important to calculate both absolute and relative inequality measures because they provide fundamentally different types of information [12]. The standard errors of these inequality measures were computed using the Taylor Series method, as implemented in HD*Calc. Additional details regarding these measures can be found in the HD*Calc documentation [8] and in related publications [13,14,15,16]. More information about the advantages and disadvantages of using different health inequality measures can be found elsewhere [12,17].

After computing inequality measures separately for each sex, we evaluated their differences between sexes by performing hypothesis testing using Z-tests to answer two different types of research questions: the first question was whether each inequality measure was different from its null value, which is 0 for absolute inequality and 1 for relative inequality. In this case, a one-sample Z-test was used. The null hypothesis was that there was no inequality present, denoted by $H_{0}:\mu =\mu_{0}$, where μ is the true value of the inequality measure and $\mu_{0}$ is its null value (0 or 1). The test statistic used was
(2)$$Z=\frac{\hat{\mu }-\mu_{0}}{SE}$$

The second question was whether there were differences in the inequality measures between males and females. In this case, a two-sample Z-test was used. The null hypothesis was that there was no difference in inequality between males and females, denoted by $H_{0}:\mu_{male}=\mu_{female}$, where $\mu_{\mathrm{male}}$ is the true value of the inequality measure for males and $\mu_{\mathrm{female}}$ is the true value of the inequality measure for females. The test statistic used was
(3)$$Z=\frac{\mu_{male}-\mu_{female}}{\sqrt{SE_{male}^{2}+SE_{female}^{2}}}$$

For both situations, we calculated and reported the corresponding *p*-values.

## 3. Results

A total of 17 countries were included in the analysis, comprising Argentina, Belize, Brazil, Canada, Chile, Colombia, Ecuador, Guatemala, Mexico, Nicaragua, Panama, Paraguay, Peru, Puerto Rico, Saint Vincent and the Grenadines, the United States of America, and Uruguay. These countries accounted for 87.9% of the population aged 10 to 24 years in the Americas in 2015 [18].

Table 1 shows the corresponding age-adjusted mortality rates due to the three causes of violent death studied, by sex and country, along with the reference population and the income per capita estimates for the baseline year 2015. Among young males, mortality rates due to road traffic injury ranged from 42.5 per 100,000 in Paraguay to 3.5 in Peru; those due to homicide from 104.9 per 100,000 in Colombia to 1.7 in Canada; and those due to suicide from 19.4 per 100,000 in Uruguay to 1.9 in Peru. Among young females, mortality rates due to road traffic injury ranged from 16.3 per 100,000 in Paraguay to 1.1 in Peru; those due to homicide from 13.9 per 100,000 in Saint Vincent and the Grenadines to 0.5 in Chile; and those due to suicide from 9.7 per 100,000 in Paraguay to 0.5 in Puerto Rico.

Regional cross-country income-related inequalities in mortality rates were conspicuous in all three causes of violent death and in both sexes. Among young males, all 11 summary measures of inequality in mortality due to road traffic injury, homicide, and suicide were statistically significantly different from their null value of no inequality, whereas among young females statistical significance was reached for all measures of inequality in mortality due to road traffic injury, nine measures of inequality in mortality due to homicide (except RD and RR), and four measures of inequality in mortality due to suicide (RD, BGV, MLD, and T) (details shown in Appendix A).

Regional cross-country sex-related differences in those income-related inequalities in mortality rates were also conspicuous in all three causes of violent death for most of the summary measures explored. Table 2 shows the results of these inequalities between young males and females. Twenty-six out of all thirty-three pairwise comparisons were statistically significant, mostly in mortality due to homicide and road traffic injury. Invariably, all these statistically significant differences between sexes showed excess mortality among young males.

The presence of mortality gradients across the income hierarchy between countries was apparent and statistically significant for both sexes and the three causes of violent death explored, yet the patterns of inequality were distinctive in each case, as explained below.

Figure 1 shows the cross-country income-related inequality gradients in mortality due to road traffic injury for young people of both sexes in the Americas. Among young men, the weighted regression line shows a strong pattern of decreasing inequality with increasing income; that is, a disproportionate concentration of the outcome of interest (mortality, in our study) at the poorer end of the social hierarchy, the so-called pro-poor inequality pattern. This pattern corresponds to the non-zero negative value of the slope index of inequality (SII), which amounts to −17.89 (95%CI: −18.87; −16.91). In contrast, there is a weak pro-rich inequality pattern among young women (SII: 1.27; 95%CI: 0.74; 1.80); that is, a disproportionate concentration of deaths due to road traffic injury at the richer end of the female social hierarchy.

Figure 2 shows the cross-country income-related inequality gradients in mortality due to homicide for young people of both sexes in the Americas. There is a pro-poor inequality pattern for both sexes, and both are statistically significant, although the disproportionate concentration of deaths due to homicide at the poorer end of the social hierarchy is more apparent among young males. The SII in young men was −81.75 (95%CI: −83.00; −80.50), whereas in young women it was −5.55 (95%CI: −5.96; −5.14).

Figure 3 shows the cross-country income-related inequality gradients in mortality due to suicide for young people of both sexes in the Americas. There is a distinctive pro-rich inequality pattern among young males (SII: 7.78; 95%CI: 7.07; 8.49); the higher the relative social position defined by income per capita of a country, the higher the disproportionality in the concentration of deaths due to suicide in young men in that country. There is no statistically discernible pattern of inequality in mortality due to suicide among young women (SII: 0.09; 95%CI: −0.36; 0.54).

## 4. Discussion

Violent unintentional and intentional deaths—including those due to road traffic injury, homicide, and suicide—are among the most important causes of premature and preventable mortality, especially in young people [19,20]. Our study highlights the underlying inequality in the distribution of these preventable risks across country income levels and sexes in people aged 10 to 24 years in the Americas by 2015, the SDG timeframe baseline year, by quantifying the magnitude of such disparities through 11 standard summary measures of health (mortality) inequality.

As expected, the quantification of social inequalities in mortality with different summary measures yielded different numerical results, yet they mostly converged toward the prevailing cause-sex inequality pattern. This is the case because each metric has unique mathematical properties inherent to its definition. For instance, some of them (such as RD and RR) quantify the inequality by a single pairwise comparison, usually between unweighted extreme social groups (such as the poorest and richest income quintiles), whereas others (such as BGV and MLD) quantify the inequality across all weighted social groups; other metrics (such as ACI, RCI, SII and RII) are sensitive to the polarity of the social hierarchy; and yet others (such as BGV and T) impose greater aversion to inequality [8,9,12,17,21,22]. On the other hand, far from being a value-neutral process, the measurement of health inequalities is inevitably affected by implicit value judgments [23]. These constraints notwithstanding, as our study has shown, most of the summary measures of health inequality converge in reproducing either a pro-poor or a pro-rich inequality pattern when this is statistically present in the data. This is so either for the absolute (i.e., the first four measures listed in Table 2) and the relative (i.e., the remaining seven) inequality summary measures for all three causes of violent death studied. Absolute inequality is assessed in the additive scale, and therefore it expresses the magnitude of inequality in absolute terms; that is, in the same units of the health variable (namely, deaths per 100,000 people aged 10–24 years). Relative inequality is assessed in the multiplicative scale, and therefore it expresses the magnitude of inequality in relative terms; that is, as a mortality rate ratio. Heuristically, both absolute and relative measures of inequality summarize the difference and the ratio, respectively, of the mortality rate at the extremes of the social hierarchy as defined by the stratifier (i.e., the country average income per capita), from the most disadvantaged (poorest end) to the least disadvantaged. It is worth noting that, of the thirty-three gender gap inequalities assessed as shown in Table 2, only five lacked statistical significance; not surprisingly, four of these referred to suicide which, among the three causes of violent death analyzed, is the least frequent (i.e., lowest rates). Moreover, the five statistically non-significant results correspond to less sophisticated summary measures of inequality: range ratio (RR) only uses the information of extreme quartiles; index of disparity (IDisp) does not take into account the size of the classes (i.e., countries); and the mean log deviation (MLD), as its name implies, works in the contracted logarithmic scale.

Consistent with our findings, Kristensen et al. [24] found an inverse relationship between the municipal proportion of high-income earners and road traffic injury mortality among adolescents (16 to 20 years old) in Norway; Pirdavani et al. [25], studying the general population of Flanders, Belgium, found a similar pro-poor inequality pattern only among males; and Harper et al. [26] found an analogous inequality pattern among US adults, but across the education gradient after controlling for age, sex, and race. In Iran, an analysis of the Multiple Indicator Demographic and Health Survey among women and children under 5 years of age by Roshanfekr et al. [27] yielded a negative SII value for road traffic injury mortality, revealing the same pro-poor pattern. To the best of our knowledge, our finding of a (weak) cross-country pro-rich inequality pattern in premature mortality due to road traffic injury among young females in the Americas has not been reported previously in the literature, and demands in-depth inquiry into its true causal nature, if any, and possible underlying mechanisms.

The finding of a clear cross-country pro-poor inequality pattern in mortality due to homicide among young males and females—along with its significant gender gap—is also supported by external evidence [28,29,30,31,32], albeit in studies on either general or adult populations. Interestingly, this inequality pattern is found to be mediated by an array of psychosocial factors, such as income inequality [28,29,30], interpersonal trust and social capital [28], social disorganization [31], institutionalization of social norms [32], and even climate temperature [30]. It remains to be explored what particular psychosocial factors and vulnerabilities might be at play among adolescents and young adults in the Americas that could explain the ubiquitous inequality pattern observed elsewhere.

Our study found a statistically significant positive mortality gradient due to suicide among young males in the cross-country social hierarchy defined by income level. The association between income and suicide has been well known since the time of Émile Durkheim [33], the formal father of modern sociology, although the direction of that association is a matter of contention [20,34]. In his seminal 1897 work, Durkheim found a direct relationship between income and suicide, a finding replicated by our study, recognizing the relevance of anomie or social fragmentation in influencing suicide. The same positive gradient has been reported by Bando et al. [34] across São Paulo city’s districts and across Brazil’s microregions. In a study of young people aged 10 to 24 years from 21 Latin American and Caribbean countries using the WHO-CDC Global School Health Survey (GSHS), Elia et al. [35] found a direct relationship between income and obesity, and then a direct relationship between obesity and suicide ideation with planning, suggesting that overweight/obesity might be on the causal pathway from economic disadvantage to suicide. In another study with worldwide GSHS data, Assarson et al. [36] found a positive association of suicide ideation with gender inequality when controlling for income. Other studies have found a negative gradient or inverse relationship between income and suicide, in the Americas [20,37,38] and elsewhere [20,39,40,41], and yet other studies have described the positive gradient between suicide and income inequality rather than income level [37,42,43,44].

The copious supply of summary measures of health inequality available not only makes the decision of which metric to use difficult in principle, but also tells something about the lack of agreement on which is the best for the task at hand. A priori, the selection of a health inequality measure should be based on the study’s primary objective as much as the primary audience of interest. Depending on that, one may choose between quantitatively simpler, more intuitive gap inequality measures or more complex, precise, and sophisticated gradient inequality measures [6,9]. From a practical standpoint, it is a good strategy to explore inequalities with several measures at the same time to verify whether the results converge and the conclusions are qualitatively consistent [12,45]. When drawing conclusions, one should not only focus on the statistical significance of the results, but also pay attention to their practical, public health significance. The latter interpretation is the most important information an analyst can provide to policymakers.

Our study has several limitations. A major one is its ecological design, which prevents making any causal claim; however, its exploratory data analysis approach allows for data pattern extraction [46], which can have very useful informative purposes. In this regard, for instance, the income level considered in this work is the per capita average per country and, as such, it does not reflect the actual income level of the study subjects (i.e., females and males aged 10 to 24 years). Another major limitation is its poor granularity. Using countries as units of analysis not only hides potentially significant vulnerabilities associated with territory and spatially patterned ecosocial inequalities, but also misses a fundamental point: for promoting health equity in the Americas, addressing within-country health inequalities is a higher imperative than exposing between-countries health inequalities. Still, the cross-country approach of our study succeeded in uncovering statistically significant patterns of health inequality that can better inform the epidemiology of violent deaths among young people in the Americas. Additionally, as suggested by Wilkinson [47], it is not uncommon for patterns uncovered at lower levels of granularity to be reproduced at higher levels of granularity. A third limitation of our study is its handling of the income gradient vis à vis the gender gap; we kept invariant the distribution of income per capita across countries regardless of sex (this was so because country data series on income per capita by sex is usually unavailable). If we assume that income is more unequally distributed among women than men, then our results may have underestimated the true mortality inequality. The intersectionality of income and gender in shaping violent death inequalities, therefore, should be taken with caution.

## 5. Conclusions

Significant inequalities in violent death among young people seem to be in place across countries of the Americas, and they seem to be socially determined by both income and sex/gender. In 2015, the start point of the 2030 Agenda, premature mortality due to road traffic injury and homicide showed a pro-poor inequality pattern, whereas premature mortality due to suicide showed a pro-rich inequality pattern in the studied population. These inequalities were particularly concentrated among young males, and were dominated by homicide as the cause of violent death. In general, the ample array of available summary measures of health inequality seem to generate convergent results. These findings shed light on the epidemiology of violent death in young people of the Americas, and can inform priorities for public health action. However, further investigation is needed at the individual level to confirm inequality patterns and to explore underlying mechanisms, sex- and age-specific vulnerabilities, and gender-based drivers of such avoidable, unnecessary, and socially costly inequalities.

## Figures and Tables

**Figure 1 ijerph-20-05256-f001:**
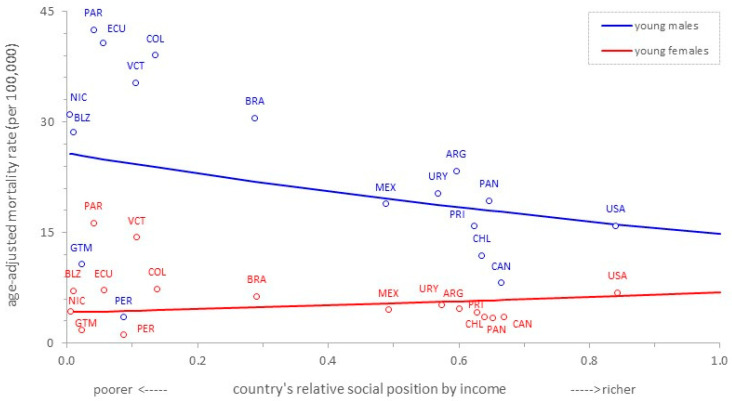
Cross-country income-related inequality gradients in mortality due to road traffic injury in young (10–24 years old) males and females. Region of the Americas, 2015.

**Figure 2 ijerph-20-05256-f002:**
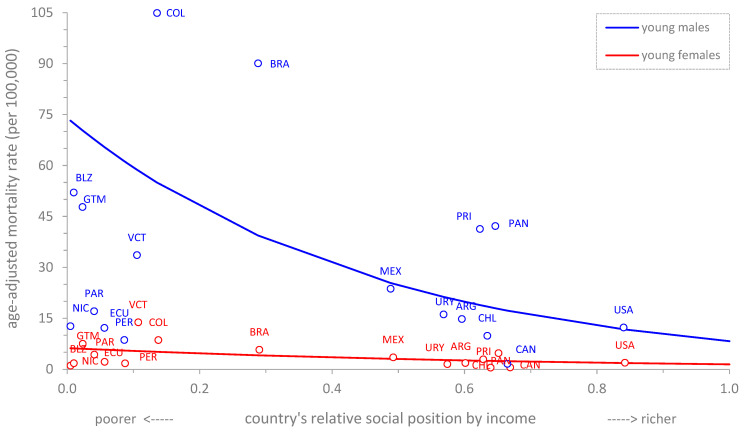
Cross-country income-related inequality gradients in mortality due to homicide in young (10–24 years old) males and females. Region of the Americas, 2015.

**Figure 3 ijerph-20-05256-f003:**
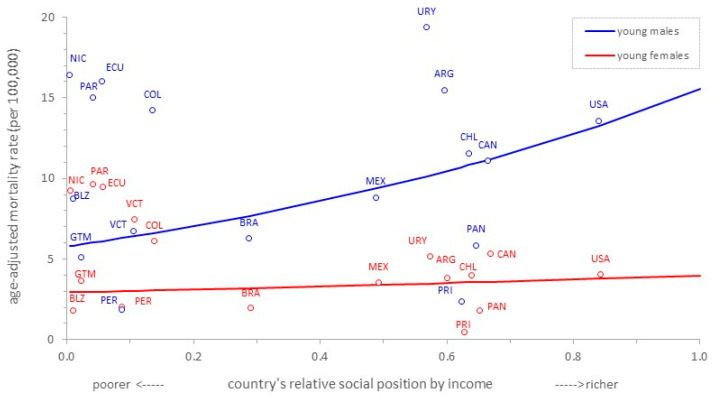
Cross-country income-related inequality gradients in mortality due to suicide in young (10–24 years old) males and females. Region of the Americas, 2015.

**Table 1 ijerph-20-05256-t001:** Age-adjusted mortality rates (per 100,000 pop) due to road traffic injury, homicide, and suicide in young (10–24 years old) males and females, reference population and average income per capita. Selected countries of the Americas, 2015.

Country	Population Aged 10–24 Years		Income per Capita	Road Traffic Injury		Homicide		Suicide	
	Male	Female		Male	Female	Male	Female	Male	Female
Argentina	5,374,733	5,128,857	23,507	23.36	4.64	14.85	1.90	15.48	3.85
				(0.66)	(0.30)	(0.53)	(0.19)	(0.54)	(0.27)
Belize	56,875	55,983	6894	28.64	7.12	52.09	1.82	8.74	1.82
				(7.16)	(3.56)	(9.67)	(1.82)	(3.91)	(1.82)
Brazil	25,571,355	24,805,494	14,749	30.52	6.38	90.10	5.80	6.27	1.99
				(0.34)	(0.16)	(0.58)	(0.15)	(0.15)	(0.09)
Canada	3,300,177	3,098,280	46,418	8.17	3.52	1.67	0.60	11.09	5.31
				(0.48)	(0.33)	(0.21)	(0.13)	(0.56)	(0.40)
Chile	2,009,299	1,930,584	23,751	11.93	3.55	9.91	0.54	11.57	3.98
				(0.75)	(0.42)	(0.68)	(0.16)	(0.74)	(0.45)
Colombia	6,343,117	6,154,428	13,914	39.05	7.34	104.92	8.66	14.23	6.11
				(0.78)	(0.35)	(1.28)	(0.38)	(0.47)	(0.32)
Ecuador	2,313,463	2,230,806	11,686	40.69	7.17	12.23	2.24	16.02	9.45
				(1.34)	(0.57)	(0.74)	(0.32)	(0.84)	(0.66)
Guatemala	2,711,234	2,652,956	7933	10.71	1.75	47.81	7.57	5.10	3.63
				(0.64)	(0.26)	(1.35)	(0.54)	(0.44)	(0.37)
Mexico	16,365,633	16,225,268	18,778	18.91	4.56	23.74	3.57	8.81	3.52
				(0.33)	(0.16)	(0.37)	(0.14)	(0.22)	(0.14)
Nicaragua	994,524	978,171	5425	31.05	4.34	12.72	1.13	16.39	9.25
				(1.83)	(0.70)	(1.17)	(0.36)	(1.33)	(1.03)
Panama	516,601	497,620	26,600	19.31	3.42	42.20	4.80	5.86	1.82
				(1.94)	(0.83)	(2.87)	(0.98)	(1.07)	(0.61)
Paraguay	943,123	906,729	11,395	42.54	16.32	17.15	4.36	15.00	9.65
				(2.04)	(1.29)	(1.29)	(0.66)	(1.21)	(0.99)
Peru	3,992,098	4,046,778	11,706	3.52	1.13	8.67	1.81	1.89	2.03
				(0.28)	(0.17)	(0.45)	(0.21)	(0.21)	(0.22)
Puerto Rico	346,172	341,849	23,570	15.86	4.17	41.37	2.96	2.36	0.49
				(1.92)	(1.01)	(3.10)	(0.86)	(0.75)	(0.34)
Saint Vincent and the Grenadines	13,362	12,724	13,047	35.31	14.38	33.65	13.89	6.73	7.44
				(15.80)	(10.18)	(15.05)	(9.82)	(6.73)	(7.44)
United States of America	33,455,413	32,044,540	59,683	15.91	6.87	12.34	1.97	13.58	4.04
				(0.22)	(0.15)	(0.19)	(0.08)	(0.20)	(0.11)
Uruguay	389,585	374,181	21,439	20.36	5.23	16.20	1.55	19.38	5.18
				(2.25)	(1.17)	(2.01)	(0.63)	(2.19)	(1.16)

**Table 2 ijerph-20-05256-t002:** Magnitude of mortality inequality, its standard error, and *p*-value, by type of inequality measurement, sex, and cause of violent death in young (10–24 years old) people. Region of the Americas, 2015.

Summary Measure of Mortality Inequality	Road Traffic Injury			Homicide			Suicide		
	Male	Female	*p* Value	Male	Female	*p* Value	Male	Female	*p* Value
Range Difference (RD)	39.02	15.18	<0.001 *	103.26	13.36	<0.001 *	17.49	9.17	<0.001 *
	(2.06)	(1.30)		(1.30)	(9.83)		(2.21)	(1.05)	
Between-Group Variance (BGV)	85.44	3.63	<0.001 *	1311.59	4.79	<0.001 *	15.43	2.66	<0.001 *
	(2.81)	(0.32)		(18.03)	(0.31)		(0.73)	(0.26)	
Absolute Concentration Index (ACI)	−2.83	0.20	<0.001 *	−12.94	−0.88	<0.001 *	1.23	0.01	<0.001 *
	(0.08)	(0.04)		(0.10)	(0.03)		(0.06)	(0.04)	
Slope Index of Inequality (SII)	−17.89	1.27	<0.001 *	−81.75	−5.55	<0.001 *	7.78	0.09	<0.001 *
	(0.50)	(0.27)		(0.64)	(0.21)		(0.36)	(0.23)	
Range Ratio (RR)	12.10	14.41	<0.001 *	62.93	25.83	0.08	10.27	19.90	0.50
	(1.14)	(2.39)		(8.13)	(19.86)		(1.63)	(14.23)	
Index of Disparity (IDisp)	597.30	456.30	<0.001 *	1924.30	651.10	<0.001 *	485.20	918.90	0.55
	(64.90)	(101.60)		(269.70)	(255.70)		(70.50)	(728.40)	
Mean Log Deviation (MLD)	0.11	0.07	<0.001 *	0.47	0.19	<0.001 *	0.10	0.09	0.16
	(0.004)	(0.006)		(0.006)	(0.01)		(0.005)	(0.007)	
Theil Index (T)	0.09	0.06	<0.001 *	0.40	0.17	<0.001 *	0.08	0.09	0.36
	(0.003)	(0.004)		(0.004)	(0.009)		(0.003)	(0.007)	
Relative Concentration Index (RCI)	−0.13	0.04	<0.001 *	−0.33	−0.24	<0.001 *	0.12	0.004	<0.001 *
	(0.004)	(0.007)		(0.002)	(0.008)		(0.005)	(0.01)	
Relative Index of Inequality (RII)	−0.82	0.22	<0.001 *	−2.07	−1.51	<0.001 *	0.75	0.02	<0.001 *
	(0.02)	(0.05)		(0.01)	(0.05)		(0.03)	(0.06)	
Kunst Mackenbach Relative Index (KMI)	1.25	0.42	<0.001 *	−0.02	0.14	<0.001 *	2.20	1.02	<0.001 *
	(0.06)	(0.01)		(0.003)	(0.01)		(0.09)	(0.06)	

* Statistically significant difference between sexes.

## Data Availability

All data and associated materials can be made available for review upon reasonable request to the corresponding author (sanhueza@paho.org).

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
