# Peer review of "Inequalities in Violent Death across Income Levels among Young Males and Females in Countries of the Americas"

_ijerph, 2023, doi:10.3390/ijerph20075256_

Round 1
Reviewer 1 Report
I read the proposed article with great interest. The authors examined inequalities in violent death among young people in countries across the Americas based on their income levels. The article is well written, the materials and methods appropriate, and the results interesting. The ecological cross-sectional study has numerous limitations of which the authors are aware. Also, the income level considered by the authors is the average per country and obviously does not reflect the people participating in the study. This aspect could be well clarified given the limitations of the study. However, the main problem with the study is that the data is from 2015. After 7 years, an update of the data would therefore be appropriate to present results that are truly representative of the current situation. Nonetheless, the evidence reported in the manuscript still helps shed light on such an important phenomenon as suicide. It is precisely for this reason that I advise the authors to consider updating the data in a future publication, as this would benefit the scientific community.
Author Response
Reviewer 1:
I read the proposed article with great interest. The authors examined inequalities in violent death among young people in countries across the Americas based on their income levels. The article is well written, the materials and methods appropriate, and the results interesting.
Thank you very much for your constructive comments –all well received.
The ecological cross-sectional study has numerous limitations of which the authors are aware. Also, the income level considered by the authors is the average per country and obviously does not reflect the people participating in the study. This aspect could be well clarified given the limitations of the study.
We concur with the reviewer. In the manuscript’s Discussion section, when referring to the ecological design as a study’s limitation, we have clarified the point raised by the reviewer by adding the following sentence: “In this regard, for instance, the income level considered in this work is the per capita average per country and, as such, it does not reflect the actual income level of the study subjects (i.e., females and males aged 10 to 24 years)”.
However, the main problem with the study is that the data is from 2015. After 7 years, an update of the data would therefore be appropriate to present results that are truly representative of the current situation. Nonetheless, the evidence reported in the manuscript still helps shed light on such an important phenomenon as suicide. It is precisely for this reason that I advise the authors to consider updating the data in a future publication, as this would benefit the scientific community.
Our study aim was to establish a SDG baseline for violent death among young people in the region of the Americas –hence to present results representative of the situation in 2015– with a explicit focus on cross-country inequalities to generate accountability on the commitment to leave no one behind. This study objective was presented in the Introduction section of the manuscript; however, taking into account the reviewer’s remarks, we have introduced some editing there to make it more clear. On a different note, we do have planned another study with more recent data to assess changes over time in inequality as well as the covid-19 impact on the distributive inequality of those health outcomes. Thanks for the advice.
Reviewer 2 Report
This manuscript is understandable and worthwhile. The association of deaths with income and gender is clearly interesting and the effects seem largely congruent across countries.
One issue that is of some concern is the 11 measures of inequality presented in Table 2. The overall conclusion is that there is excess mortality among males and it is unclear to me why all eleven indices are necessary or useful. Some explanation about this issue would be helpful.
A sentence on page 5 last line of second paragraph should use the word showed rather than shown.
The pro rich mortality pattern among women's mortality due to road traffic would seem due in part simply to the availability of vehicles in places like Canada and the US yet that pattern did not exist for males. I would like to see more discussion of the authors interpretation of this pro rich mortality pattern,
Author Response
Reviewer 2:
This manuscript is understandable and worthwhile. The association of deaths with income and gender is clearly interesting and the effects seem largely congruent across countries.
Thank you very much for your constructive comments –all well received.
One issue that is of some concern is the 11 measures of inequality presented in Table 2. The overall conclusion is that there is excess mortality among males and it is unclear to me why all eleven indices are necessary or useful. Some explanation about this issue would be helpful.
The decision to report our results through 11 summary measures of inequality was strictly operational: they constitute a standard set of measurements that comes with the computational analytical tool used: HD*Calc, a well-known, well-documented, freely available software produced by the US National Cancer Institute –we did want to maximize the reproducibility of results and avoid cherry-picking some measures over others. The fact that HD*Calc computes 11 different summary measures of inequality reflects to a certain extent a lack of agreement among social epidemiologists and other researchers on a gold standard, and a recognition of the different technical subtleties and attributes of each of them (such as, for instance, the scale of measurement, weighting, degree of aversion to inequality, etc.). A meaningful discussion of these specific issues, we believe, falls outside the scope of our paper. However, we did refer the concerned reader to the relevant references available. In addition, we devoted two full paragraphs in the Discussion section (lines 248-260, and lines 333-347) to the issue of the copious supply of summary measures of inequality, their unique properties and how to better guide a selection decision. In reference to the overall conclusion, we beg to differ with the reviewer’s: the overall conclusion is that there are distinctive cross-country inequality gradients in the three outcomes explored, both in young men and young women.
A sentence on page 5 last line of second paragraph should use the word showed rather than shown.
Done. Thanks.
The pro rich mortality pattern among women's mortality due to road traffic would seem due in part simply to the availability of vehicles in places like Canada and the US yet that pattern did not exist for males. I would like to see more discussion of the authors interpretation of this pro rich mortality pattern.
The apparent dissociation in the road traffic injury mortality inequality pattern between young males and females shown in Figure 1 caught our attention as well; we re-run the analysis and reproduced the results presented: the strong pro-poor pattern among young males and the weak pro-rich pattern among young females was captured by all 5 complex measures of gradient inequality (i.e., ACI, SII, RCI, RII, and KMI). Incidentally, we found a typo in Table 2 –and again in Supplementary Table 1– which has been corrected. We have made a few editing in both the respective Results and Discussion sections to make this clearer. As a matter of fact, we devoted a whole paragraph in the Discussion section (lines 264-284) to better contextualize our findings, and made some editorial changes to improve it. In our exploratory analyses, we did not take into account regressors such as the country’s average road motor vehicle fleet size or the average amount of distance traveled by road motor vehicles which, as pointed out by the reviewer, may confound the relationship between relative social position and risk of dying due to road traffic injuries among sexes (yet hardly explain it, though). Due to the ecological design of our study, we did refrain from providing further speculative explanation to this. We did, however, highlight the need for further in-depth causal research into this topic.
Reviewer 3 Report
This manuscript explores the underlying inequality in the distribution of violent deaths (i.e., road traffic injury, homicide, and suicide) across country income levels and sexes in people aged 10 to 24 in the Americas by 2015. The results are interesting. However, I felt that there were a number of issues that the authors need to address, which I will outline below. With these, I recommend rejection of the manuscript in its current form, but a re-submission may be encouraged.
1.I suggest that add the specific process of selecting 17 countries out of 48 countries for this study.
2. There are many factors that contribute to violent deaths, so why focus on gender and income?
3. Violent deaths (i.e., those due to road traffic injury, homicide, and suicide) are among the most important causes of premature and preventable mortality in young people.” Is there any support data? “These 139 countries accounted for 87.9% of the population aged 10 to 24 years in the 140 Americas in 2015” in Results Section. Where are this data from?
4. Why target people aged 10-24 years?
5. It's been seven years since 2015, so why not use the latest data?
6.What’s the specific different information between absolute and relative inequality measures? In Materials and Methods section, the importance of calculating the absolute and relative inequality measures is mentioned, but in Results and Discussion sections, there is no mention at all of what these two kinds of figures mean.
7.About table 1, I suggest adding the standard error of age-adjusted mortality rates due to the three causes of violent death studied, by sex and country. About table 2, there are seven p values that are not significant, but are not explained in Discussion section.
8. Some words are not accurate. For example, in the sentence “Among young men, the weighted regression line shows a pattern of decreasing inequality with increasing income”, the “inequality” should be “mortality rate” because the Y-axis is age-adjusted mortality rate, not mortality inequality. The full ms need to be checked.
9. What’s the statistical method of mortality gradients between countries for both sexes and the three causes of violent death? More information needed.
10. I would also suggest a more detailed explanation of the findings and appropriate addition of theoretical support.
Author Response
Reviewer 3:
This manuscript explores the underlying inequality in the distribution of violent deaths (i.e., road traffic injury, homicide, and suicide) across country income levels and sexes in people aged 10 to 24 in the Americas by 2015. The results are interesting. However, I felt that there were a number of issues that the authors need to address, which I will outline below. With these, I recommend rejection of the manuscript in its current form, but a re-submission may be encouraged.
Thank you very much for your constructive comments –all well received.
- I suggest that add the specific process of selecting 17 countries out of 48 countries for this study.
We have reviewed the corresponding paragraph in the Material and Methods section and made some relevant clarifications. As a matter of fact, there were not 48 but 34 countries initially considered for our study –based on availability of 2015 data in the regional mortality database by the end of year 2019, when we started this study. The specific, and standard, procedure followed to finally select the 17 countries is published elsewhere, and it was appropriately referenced.
- There are many factors that contribute to violent deaths, so why focus on gender and income?
Well, why not? The subject matter of the study is a prerogative of the research team, isn’t it? We thank the reviewer, though, for pointing out the need to make it explicit the rationale behind the study design. We chose income as a social stratifier given its particular relevance in the Region of the Americas as the main driver of social inequalities in health –the Americas being regarded as the most inequitable region of the world based on the magnitude of its Gini coefficient over time. On the other hand, gender is a cross-cutting theme of extreme regional (and worldwide) relevance, and we tried to get it reflected by further disaggregating the data –and the analysis– by sex. Target 17.8 of the Sustainable Development Agenda to 2030 calls indeed for such income-, gender-, and age- disaggregation. We have made some adjustments in the narrative of the Introduction section to make it more visible the answer to the question posed by the reviewer.
- “Violent deaths (i.e., those due to road traffic injury, homicide, and suicide) are among the most important causes of premature and preventable mortality in young people.” Is there any support data?
Thanks for bringing this point to our attention. The quoted phrase comes from our abstract, which cannot contain references. We involuntarily omitted to invoke it again in the Introduction and provide supporting reference. This involuntary omission has been fixed now.
“These 139 countries accounted for 87.9% of the population aged 10 to 24 years in the 140 Americas in 2015” in Results Section. Where are this data from?
This straightforward computation was made by us, using the most recent revision of the United Nations Population Division estimates (World Population Prospects: The 2022 Revision). More specifically, from its file POP/DB/WPP/Rev.2022/POP/F01-1, there were 206’182,005 people of both sexes aged 10 to 24 in 2015 (midyear), out of a total of 234’587,136 in the Region of the Americas; that is, 87.89%. We have added the corresponding reference in the Results section.
- Why target people aged 10-24 years?
Again, we respectfully consider this our prerrogative as a research team. We are deeply concerned about the apparent invisibility of the young population vis-à-vis other groups across the life course in the social epidemiology and public health research agenda in our region –not to mention in the literature on health inequalities, as well as the disproportionately share of premature deaths due to both unintentional and intentional violent causes in this population group. Our study also aims, humbly, to raise awareness about this unwanted situation.
- It's been seven years since 2015, so why not use the latest data?
Our study aim was to establish a SDG baseline for violent death among young people in the region of the Americas –hence to present results representative of the situation in 2015– with a explicit focus on cross-country inequalities to generate accountability on the commitment to leave no one behind. This study objective was presented in the Introduction section of the manuscript; however, taking into account the reviewer’s query, we have introduced some editing there to make it more clear. On a different note, we do have planned another study with more recent data to assess changes over time in inequality as well as the covid-19 impact on the distributive inequality of those health outcomes.
6.What’s the specific different information between absolute and relative inequality measures? In Materials and Methods section, the importance of calculating the absolute and relative inequality measures is mentioned, but in Results and Discussion sections, there is no mention at all of what these two kinds of figures mean.
We thank you for rising this point. We have revised the manuscript and added a whole paragraph at the beginning of the Discussion section to take care of this involuntary, yet relevant omission.
7.About table 1, I suggest adding the standard error of age-adjusted mortality rates due to the three causes of violent death studied, by sex and country. About table 2, there are seven p values that are not significant, but are not explained in Discussion section.
We have modified Table 1 and added the standard errors for each age-adjusted, cause-, sex-, and country-specific mortality rate estimate as requested by the reviewer. We have also added a paragraph (right after the one added in response to reviewer’s point 6, above) discussing about the statistically non-significant gender gap inequality results presented in Table 2 (which are only 5 out of the 33).
- Some words are not accurate. For example, in the sentence “Among young men, the weighted regression line shows a pattern of decreasing inequality with increasing income”, the “inequality” should be “mortality rate” because the Y-axis is age-adjusted mortality rate, not mortality inequality. The full ms need to be checked.
We beg to differ with the reviewer’s comment. Although it is a literally correct interpretation vis-à-vis the graphical array (i.e., the relative social position scale as the abscissa or x-coordinate, the age-adjusted mortality rate as the ordinate or y-coordinate), we are using standard parlance to describe patterns of inequality in the presence of health (or mortality) gradients across a social (or socioeconomic) hierarchy, as recommended by the World Health Organization (see reference 11, for instance) following the seminal work of Prof. Cesar Victora et al (Lancet 2005;366:1460-6). In Figure 1 of our manuscript, for instance, the inverse gradient of mortality due to road traffic injury in young men across the income-defined relative social hierarchy means that the difference in mortality between the highest end of said social hierarchy and any other point of such hierarchy is maximum at its lowest end; therefore, inequality decreases as social position increases. As suggested by the reviewer, we have checked the entire Results and Discussion sections of the paper and found our narrative explicit enough to convey the right message; for example, in the same sentence quoted by the reviewer, it can be read immediately after: “that is, a disproportionate concentration of the outcome of interest (that is, mortality –in our study) at the poorer end of the social hierarchy, the so-called pro-poor inequality pattern.”
- What’s the statistical method of mortality gradients between countries for both sexes and the three causes of violent death? More information needed.
We are not quite sure to understand the question posed by the reviewer. Please refer to our reply to the previous question (above). In our manuscript we have provided plenty of authoritative references about the more methodological and statistical aspects of building summary measures of health inequality for the concerned reader, considering that an in-depth discussion of these themes are beyond the scope of our study.
- I would also suggest a more detailed explanation of the findings and appropriate addition of theoretical support.
We thank the reviewer for this suggestion. To the best of our abilities, we have made several changes and editing in the revised manuscript in order to address these issues.